# Performance estimation of two in-house ELISA assays for COVID-19 surveillance through the combined detection of anti-SARS-CoV-2 IgA, IgM, and IgG immunoglobulin isotypes

**Alfredo Ramírez-Reveco**[1☉], **Gerardo Velásquez**[1,2‡], **Christopher Aros**[1,3‡], **Gabriela Navarrete**[1,3‡], **Franz Villarroel-Espíndola**[4], **Maritza Navarrete**[2], **Alberto Fica**[5], **Anita Plaza**[6], **Natalia Castro**[6], **Claudio Verdugo**[6,7], **Gerardo Acosta-Jamett**[6,8], **Cristóbal C. Verdugo**[6,8☉]*

1 Instituto de Ciencia Animal, Facultad de Ciencias Veterinarias, Universidad Austral de Chile, Valdivia, Chile, 2 Laboratorio de Biología Molecular, Hospital Base de Valdivia (HBV), Valdivia, Chile, 3 Escuela de Bioquímica, Facultad de Ciencias, Universidad Austral de Chile, Valdivia, Chile, 4 Laboratorio Medicina Traslacional, Instituto Oncológico Fundación Arturo López Pérez, Santiago, Chile, 5 Sub Departamento de Medicina, Hospital Base Valdivia; Instituto Medicina, Facultad de Medicina, Universidad Austral de Chile, Valdivia, Chile, 6 Center for the Surveillance and Evolution of Infectious Diseases (CSEID), Universidad Austral de Chile, Valdivia, Chile, 7 Instituto de Patología Animal, Facultad de Ciencias Veterinarias, Universidad Austral de Chile, Valdivia, Chile, 8 Instituto de Medicina Preventiva, Facultad de Ciencias Veterinarias, Universidad Austral de Chile, Valdivia, Chile

☉ These authors contributed equally to this work.
‡ GV, CA and GN also contributed equally to this work.
* cristobal.verdugo@uach.cl

## Abstract

The main objective of this study was to estimate the performance, under local epidemiological conditions, of two in-house ELISA assays for the combined detection of anti-SARS-CoV-2 IgA, IgM, and IgG immunoglobulins. A total of 94 serum samples were used for the assessment, where 44 corresponded to sera collected before the pandemic (free of SARS-CoV-2 antibodies), and 50 sera were collected from confirmed COVID-19 patients admitted to the main public hospital in the city of Valdivia, southern Chile. The Nucleocapsid (Np) and the receptor-binding domain (RBD) proteins were separately used as antigens (Np and RBD ELISA, respectively) to assess their diagnostic performance. A receiver operating characteristic (ROC) analysis was performed to estimate the optical density (OD) cut-off that maximized the sensitivity (Se) and specificity (Sp) of the ELISA assays. Np ELISA had a mean Se of 94% (95% CI = 83.5–98.8%) and a mean Sp of 100% (95% CI = 92.0–100%), with an OD 450 nm positive cut-off value of 0.88. On the other hand, RBD ELISA presented a mean Se of 96% (95% CI = 86.3–99.5%) and a mean Sp of 90% (95% CI = 78.3–97.5%), with an OD 450 nm positive cut off value of 0.996. Non-significant differences were observed between the Se distributions of Np and RBD ELISAs, but the latter presented a significant lower Sp than Np ELISA. In parallel, collected sera were also analyzed using a commercial lateral flow chromatographic immunoassay (LFCI), to compare the performance

**Data Availability Statement:** All relevant data are within the paper and its Supporting information files.

**Funding:** - CrV - ANID COVID 0585 - Agencia Nacional de Investigacion y Desarrollo del Ministerio de Ciencias de Chile - www.anid.cl - The funders had no role in study design, data collection and analysis, decision to publish, or preparation of the manuscript.

**Competing interests:** The authors have declared that no competing interests exist.

of the in-house ELISA assays against a commercial test. The LFCI had a mean sensitivity of 94% (95% CI = 87.4–100%) and a mean specificity of 100% (95% CI = 100–100%). When compared to Np ELISA, non-significant differences were observed on the performance distributions. Conversely, RBD ELISA had a significant lower Sp than the LFCI. Although, Np ELISA presented a similar performance to the commercial test, this was 2.5 times cheaper than the LFCI assay (labor cost not considered). Thus, the in-house Np ELISA could be a suitable alternative tool, in resource limited environments, for the surveillance of SARS-CoV-2 infection, supporting further epidemiological studies.

## Introduction

The emergence of the severe acute respiratory syndrome coronavirus 2 (SARS-CoV-2), the etiological agent of COVID-19, has caused a global pandemic, which 2.5 years after the first reported case (December 2019) has been linked to more than 5 million deaths and more than two hundred million cumulative cases [1]. In Chile, a country of more than 19 million people, the first case was detected in March 2020, and 36 months after that case more than 1.6 million people have been diagnosed and more that 37 thousand deaths have been associated to this epidemic [2]. Government response was characterized by national level lock downs together with contact tracing and isolation of diagnosed cases, where infected people was detected mainly through passive surveillance of suspected cases [3]. In general, since its emergence, several serological assays have been developed to detect the presence of specific antibodies against SARS-CoV-2 [4]. Although, serological approaches cannot distinguish between acute and chronic infection, these types of tests are useful for *i*) the identification of individuals who have developed an immune response, *ii*) aid in contact tracing, *iii*) monitoring infection dynamic in the general population, and *iv*) the development of clinical trials [5, 6].

The immunoglobulin time response is between 4 to 10–14 days after the onset of symptoms, which limits its applicability for the diagnosis during the acute phase [7, 8]. However, the IgM and IgA anti SARS-CoV-2 antibodies develop rapidly in response to the infection, and their detection can significantly increase the diagnostic sensitivity of SARS-CoV-2 infection, when serological tests are combined with molecular tests [9]. In particular, IgA antibodies play an important role in mucosal immunity, where IgA may be a better marker of early infection than IgM [10–12]. However, most COVID-19 serological tests are based on the detection of IgM and/or IgG antibodies [13]. The preference of IgG and/or IgM detection over IgA, probably is related to a lower specificity of this immunoglobin despite an earlier onset in comparison with IgG and IgM [11]. Nevertheless, the use of assays detecting IgA, along with IgG and IgM, may be useful in scenarios where it is necessary to maximize the diagnostic sensitivity of the test, such as a screening tool for a surveillance program, for example. Additionally, IgA assays may also be helpful in patients with atypical symptoms, in asymptomatic cases, or when Reverse transcription-quantitative polymerase chain reaction (RT-qPCR) results remains negative in suspected subjects [14, 15].

The detection of circulating antibodies against SARS-CoV-2, as part of a surveillance program, requires the use of tools with known sensitivity and specificity [16]. Those parameters would allow to estimate key epidemiological variables, such as the true prevalence (TP), when the assay is used in randomized studies. Moreover, accurate TP estimates could be used to assess the performance of passive surveillance systems that most countries have implemented as part of their COVID-19 control strategies [17, 18].

The main serologic assays used for SARS-CoV-2 detection include the lateral flow chromatographic immunoassay (LFCI) and ELISA tests. In these kinds of assays, the most used viral proteins as antigens are the nucleocapsid protein (Np), which plays a role in the transcription and replication of the virus [19, 20] and the subunits S1 and S2 of the spike (S) protein [21]. Specifically, S1, containing the receptor binding domain (RBD) for the host angiotensin-converting enzyme (ACE2) receptor; and the S2, containing elements needed for membrane fusion [22, 23]. Previous evidence has suggested that the IgG antibodies aiming for the S protein are more specific than the anti-Np protein [24, 25]. On the other hand, the IgG aiming Np may be more sensitive than those anti-S proteins, particularly in the early phase of infection [24, 25]. This could be explained by the relatively high homology in aminoacidic sequence of the SARS-CoV-2 protein Np with the nucleocapsid proteins of other Coronaviridae and other viruses [24, 26]. Thus, the increased sensitivity of the anti-Np antibody response detection could be at the expense of specificity. The latter may be due to the potential cross-reaction of serological tests to other similar viruses circulating in the target population, increasing the false positive rate. In consequence, the performance of any serological assay, such as ELISA tests must be optimized and validated under local conditions, accounting for endemically circulating viruses.

The objective of this study was to evaluate and validate two in-house ELISA assays for the combined detection of SARS-CoV-2 IgG, IgM and IgA antibodies. To be used for the surveillance of COVID-19 in the general population, and to support further epidemiological studies. In particular, diagnostic performance indices, such as sensitivity (Se) and specificity (Sp), were estimated using sera from pre-pandemic individuals and confirmed COVID-19 patients.

## Material and methods

### Study population and sample collection

The present study was developed following the Standards for the Reporting of Diagnostic Accuracy Studies (STARD) guidelines proposed by Cohen et al. [27]. Sample size was calculated using the equation proposed by Obuchowski, NA [30] for conducting a receiver operating characteristic (ROC) analysis, to estimate test optical density (OD) cut-off, sensitivity, and specificity (further details in the statistical section). Based on this methodology, a minimum of 43 infected and 43 uninfected samples was needed to yield a study power of 80%, a significance level of 5%, and an expected area under the curve (AUC) of 65%. In the present study, a total of 94 individuals were enrolled, where 50 corresponded to confirmed cases of SARS-CoV-2 infection (24 women and 26 men), and 44 corresponded to sera collected between June and July 2019, thus regarded as free of SARS-CoV-2 antibodies sera (pre-pandemic samples). All 50 confirmed cases corresponded to unvaccinated patients admitted to the Hospital Base de Valdivia (HBV), Valdivia, southern Chile, from April to November 2020, and they were confirmed by a standard RT-qPCR assay using nasopharyngeal swab samples, as previously described [28]. At the time of sera collection at the HBV, all were symptomatic cases, presenting different degrees of COVID-19 complications, from very mild to severe, which eventually required hospitalization at the HBV. In this line, confirmed cases presented a median of 4 days (interquartile range (IQR): 2–6 days) between symptoms onset and RT-qPCR diagnosis, whereas serum samples were collected with a median of 11.5 days (IQR: 9–15 days) after symptoms onset. This group presented a median age of 58 years (1st Q: 52.5 years and 3rd Q: 68.5 years). On the other hand, samples from non-COVID-19 individuals were obtained from a serum bank of an epidemiological study on cystic echinococcosis, where sera were collected from the general population in the Coquimbo region, Chile (grant: EULAC/FONIS T020067). This group was composed of 27 women and 17 men, with a median age of 53.5 years (1st Q: 41.5 years and 3rd Q: 63.5 years), and 7 out of 44 control-participants presented some type of chronic disease, mainly diabetes.

**Ethics approvals.** Serum samples from non-COVID-19 individuals were obtained and handled following the protocol accepted by the Scientific Ethics Committee of the Faculty of Medicine at the Universidad Católica del Norte (CECFAMED-UCN), Coquimbo-Chile, approved under the resolution CECFAMED-UCN Nº 81/2019. The samples from the confirmed COVID-19 patients were collected following the sample collection protocol of the HBV, where 4–6 cc of blood were obtained by peripheral venipuncture using yellow or red cap tubes (BD Vacutainer®) and processed before 4 hours, where samples were centrifuged for 10 min at 1000×g and the obtained sera was stored at -80˚C until use. The same sample collection protocol was used for non-COVID-19 individuals. The authorization for the use of anonymized-stored sera (EULAC/FONIS T020067) for COVID-19 research purposes, was granted by the Scientific Ethics Committee of the Servicio de Salud Valdivia (SSV), Ministry of Health of Chile, under the resolution SSV Ord.N˚187/2020. In both cases (samples from non-COVID-19 individuals and from COVID-19 patients) the volunteers or patients formalized their consented intention to participate in their respective study (cystic echinococcosis and COVID-19, respectively) by signing a written informed consent form. All volunteers were over 18 years old at the time of sample collection.

## Study design and laboratory analysis

Aiming to develop a surveillance tool, with known sensitivity and specificity, for COVID-19. Two in-house ELISAs for the combined detection of specific anti-SARS-CoV-2 isotype antibodies (IgA, IgM, and IgG) were assessed. To estimate their performance (sensitivity & specificity), sera from pre-pandemic and confirmed cases were tested using Np and RBD proteins separately as antigens, hereinafter referred to as Np ELISA and RBD ELISA, respectively. In this way, performance estimates were obtained for each in-house ELISA separately. Analyzes were run in duplicate and reported OD corresponded to the mean between runs. As a quality assurance element, the coefficient of variation between runs was estimated.

**In-house ELISAs for combined detection of specific SARS-CoV-2 antibodies.** The proteins used were the following: Recombinant SARS-CoV-2 Np protein (Met1-Ala419, with a C-terminal 6-His tag) from *Spodoptera frugiperda* (R&D System, Catalog Number 10474.CV) (accession # YP_009724397.2). Recombinant SARS-CoV-2 S1 subunit protein (Arg319-Phe541, with a C-terminal 6-His tag) of Host Cell Receptor Binding Domain (RBD) from HEK293 cells (Raybiotech, Catalog Number: 230–30162) (accession # QHD43416.1). For coating step, a final volume of 50 μl of 20 ng Np or RBD proteins [29] were seeded in a 96-well plate in carbonate buffer pH 9.6 per well and incubated overnight at 4˚C. Subsequently, 3 washes of 5 minutes were carried out with 1x Phosphate Buffered Saline (PBS), Tween20 0.05% and the wells were blocked with 200 μl of 1x PBS, 5% of Bovine Serum Albumin (BSA), 0.05% Tween20 for 2 hours at 37˚ C, and the content of the plate was discarded, and the wells were washing three times with 1x PBS), Tween20 0.05%. For the serum samples loading, a final volume of 50 microliters of sample (1:40 dilution) in 1x PBS buffer 0.1% w/v of BSA; 0.05% v/v Tween20 [29, 30] was applied in the wells and incubated for 2 hours at 37˚ C. After the incubation time, 3 washes of 5 minutes were carried out with 1x PBS, 0.05% Tween20. For antibodies hybridization step, 100 μl per well of Anti-Human IgA/IgG/IgM (H&L) goat polyclonal antibody (HRP) (Rockland, R.609-103-130) at a dilution of 1: 10,000 in 1x PBS, 0.1% BSA-0.05% Tween20, were added and incubated for 1 hour at 37˚ C, the content of the plate was discarded, and the wells were washes three times of 5 minutes with 1x PBS, 0.05% Tween20. Finally, the washed wells were developed with 3, 3', 5, 5'—Tetramethylbenzidine (TMB) reaction and read at 450 nm using a microplate reader HR801 (Shenzhen Highcreation Technology Co. Ltd). A detailed list of all reagents and instruments used has been included as S1 Table.

**Performance comparison between the in-house ELISAs and a commercial test.** To compare the performance of the in-house assays (Np & RBD ELISAs) against a commercially available test. The collected sera were additionally tested using a LFCI kit (The Diagnostic Kit for IgM / IgG Antibody to Coronavirus (SARS-CoV-2) (Lateral Flow), LIVZON, China). Serum samples for rapid test were processed according to the manufacturer's instructions (Zhuhai Livzon Diagnostics Inc.). Briefly: 10 μL of serum sample were added into the sample well of both IgM test and IgG test cassettes, then 2 drops (about 100 μL) of sample diluent were added vertically. Finally, within 1 to 15 minutes after sample addition, the results were interpreted as positive when both the control line and the test line appeared. The results were interpretated as negative when only control line appeared.

This LFCI test was widely used by the Chilean Health Services in the early stages of the pandemic, and it can separately detect IgG and IgM immunoglobulin isotypes, using Np as antigen. For this assessment, the LFCI interpretation criteria considered that any serum showing a signal for IgG or IgM was positive to SARS-CoV-2 infection. Statistical comparison between the LFCI and in-house assays are described below. To complement the comparison between the in-house ELISAs and the commercial LFCI test performance, a cost per sample analyzed at laboratory level was included. The latter evaluation only considered the cost of the reagents (in-house ELISAs) or the commercial value of the kit (LFCI). A list with the costs in USD of all regents used for the in-house ELISA has been included as S2 Table.

## Statistical analysis

Collected data were analyzed using GraphPad Prism 6 statistical software (USA) and the R software v4.0.2 [31]. They included mean values and standard deviation (mean ± SD) of the optical density (OD) distribution. Those values were subjected to a background subtraction step before data analysis. For comparative analysis of the infected and pre-pandemic OD distributions, a Mann-Whitney U-test was used. A ROC analysis was conducted to determine the best OD cut-off for the in-house ELISA tests, in addition to the sensitivity and specificity values, for Np or RBD antigens. The ROC and AUC values were calculated by comparing infected (sera from confirmed COVID-19 patients) and non-infected specimens (sera from pre-pandemic individuals), based on a logistic regression model. The agreement between Np ELISA and RBD ELISA results were assessed using the Cohen' kappa index and the paired McNemar's test. Statistical differences in sensitivity and specificity distributions between Np and RBD ELISAs were evaluated using the Kolmogorov–Smirnov test. The overlap area between Np and RBD sensitivity and specificity curves was estimated using the "overlapping" package [32] for the R software. Finally, in-house ELISAs and LFCI dichotomic results (positive/negative) were cross tabulated in two-by-two tables and the agreement between them was also evaluated using Cohen' kappa index and the paired McNemar's test, in addition to the Kolmogorov–Smirnov test to assess statistical differences between the performance distributions of both tests, where Np ELISA and RBD ELISA were separately compared against the commercial LFCI test. Anonymized in-house ELISAs and LFCI results (raw data) have been included as S3 Table.

## Results and discussion

### In-house ELISAs for specific SARS-CoV-2 antibodies using Np or RBD antigens

The Fig 1 shows a representative scheme of used proteins and their location in the SARS-CoV-2 virus, together with the OD distributions of the Np ELISA (left) and RBD ELISA (right) for

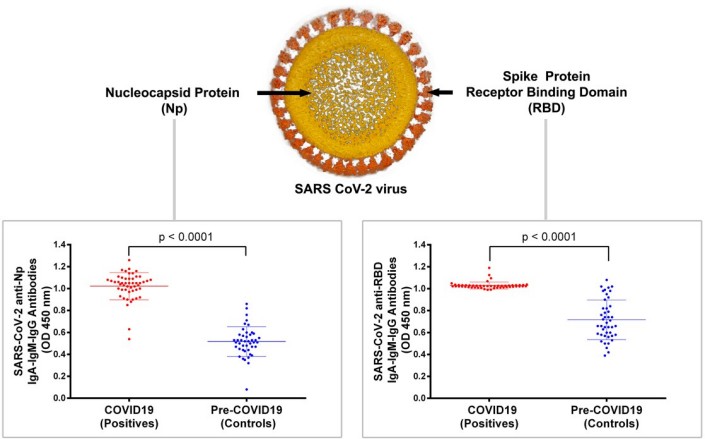

**Fig 1. Combined immunodetection of SARS-CoV-2-specific IgA/IgM/IgG immunoglobulins in response against the Nucleocapsid (Np) and receptor-binding domine of spike protein (RBD).** Oil painting picture schematic of SARS-CoV-2 viral structure (top panel), N-protein (Np) and RBD from S-protein are showed in light orange and red, respectively. Absorbance at 450 nm plotted for combined detection of IgA, IgM, and IgG specific immunoglobulin isotypes for Np and RBD ELISAs (lower left and lower right panel, respectively). Forty-four sera from pre-pandemic individuals and fifty sera from COVID-19 patients were tested. To compare both groups, a Mann-Whitney U-test was used. Mean+/-SD and P values are showed (P < 0.0001).

confirmed cases and controls. At the comparison between duplicated runs, both assays showed an adequate consistency, and a low error level, with intra-assay variation of 11.2% and 13.4% for Np and RBD ELISAs, respectively. In this way, antibodies against Np protein showed that sera from COVID-19 patients had a significantly higher (97%) absorbance than sera from pre-pandemic individuals (1.02+/-0.12 vs 0.51+/-0,13 OD, respectively) (P < 0.0001). On the other hand, although to a lesser extent, the immunoassay against the RBD protein also showed that sera from patients with COVID-19 presented a significantly higher absorbance (43.5%) than sera from pre-pandemic individuals (1.09 +/- 0.03 vs 0.71 +/- 0.18 OD, respectively) (P < 0.0001). Interestingly, sera from confirmed COVID-19 patients showed a more homogeneous distributions in their absorbances with the RBD immunoassay than with Np (Fig 1, right and left, respectively). Furthermore, these distributions suggest that most of the analyzed sera presented antibodies that recognize and bound to both proteins (Np and RBD), and only 2 patients presented antibodies only against the RBD protein.

## In-house ELISAs performance assessment (sensitivity and specificity estimation)

The ROC analysis of the Np ELISA indicated a mean sensitivity of 94% (95% CI = 83.5–98.8%) and a mean specificity of 100% (95% CI = 92.0–100%), with AUC of 0.99 (95% CI: 0.97–1.00) and an OD 450 nm positive cut-off value of 0.88 (Fig 2A). On the other hand, the ROC analysis of the RBD ELISA had a mean sensitivity of 96% (95% CI = 86.3–99.5%) and a mean specificity of 90% (95% CI = 78.3–97.5%), with AUC of 0.96 (95% CI: 0.91–1.00) and an OD 450 nm positive cut off value of 0.996 (Fig 2B). Those cut-offs represent the point in the ROC curves where the sensitivity and specificity were maximized for this set of samples.

The agreement between Np and RBD ELISAs was assessed as substantial to perfect (kappa = 0.81, 95% CI = 0.69–0.93) based on the qualitative scale interpretation proposed by Landis and Koch [33] of Cohen's kappa values. Additionally, the dichotomous ELISA results (positive/negative) were not statistically different (P = 0.18) between the two antigens. Although RBD ELISA presented a relatively higher sensitivity than Np ELISA, those

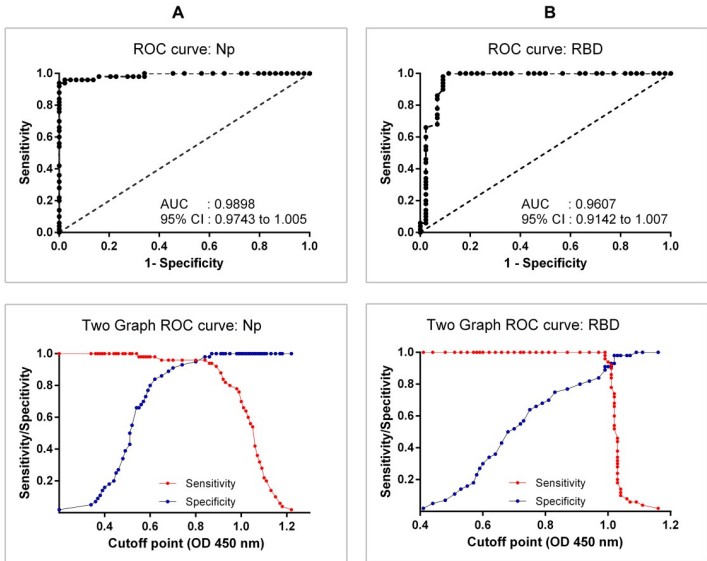

**Fig 2. Receiver operating characteristic (ROC) curves for the combined detection of SARS-CoV-2 IgA, IgM, and IgG immunoglobulin isotypes using RBD or Np antigens.** Graph of ROC curve and two graph curves (sensitivity and specificity for different cut-off) for Np-specific immunoglobulins (Panel A: top and bottom, respectively), and RBD-specific immunoglobulins (Panel B: top and bottom, respectively), are shown. Forty-four sera from pre-pandemic individuals and fifty sera from COVID-19 patients were tested. AUC, area under the curve of ROC.

differences were not statistically significant (P = 0.37), with an overlapping area between both curves of 69.7% (Fig 3A). Conversely, the use of RBD protein as antigen was associated to a statistically lower (P < 0.01) specificity, with a shared distribution of 10.8% between the Np and RBD proteins (Fig 3B).

## In-house ELISAs and commercial LFCI test comparison

The cross-tabulation between the LFCI test results and SARS-CoV-2 infection status are presented in Table 1, whereas the cross-tabulation of the dichotomous ELISA and commercial test results are shown in Table 2. Based on the infection status (gold standard), the LFCI test presented a mean sensitivity of 94% (95% CI = 87.4–100%) and a mean specificity of 100% (95% CI = 100–100%). The Cohen' kappa index showed a substantial to perfect (kappa = 0.89, 95% CI = 0.80–0.98) agreement between the Np ELISA and the LFCI test, whereas the agreement between RBD ELISA and the LFCI test was categorized as substantial kappa = 0.78, 95% CI = 0.66–0.91). Non-significant differences in the sensitivity (P = 0.999) or specificity (P = 0.999) distributions were observed for the comparison between Np ELISA and the LFCI test. Conversely, the comparison between RBD ELISA and the LFCI test render a significant difference in the specificity distribution (P < 0.01), but non-differences were observed between them for the sensitivity distributions (P = 0.999). The use of the LFCI had a cost of $10.57 USD per serum analyzed, whereas the Np ELISA and RBD ELISA had costs of $4.23 USD and $2.65 USD per serum, respectively. Thus, the in-house ELISAs were between 2.5–4.0 times cheaper than the used commercial test, when labor it is not considered.

## General discussion

The present research reports the performance estimation of two in-house ELISA assays for the surveillance of SARS-CoV-2 infection, which showed a comparable performance than a

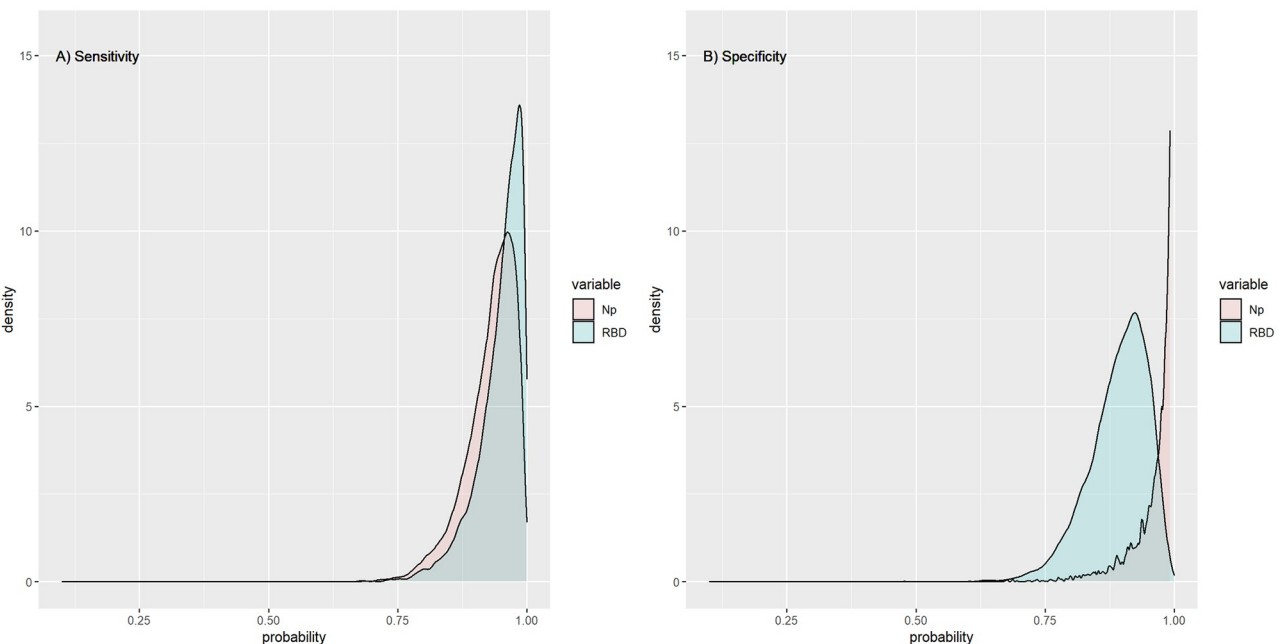

**Fig 3. Estimated sensitivity and specificity distribution curves for the selected OD cut off (Np = 0.88 & RBD = 0.996).** The overlapping area between the sensitivity curves of Np ELISA vs. RBD ELISA (Panel A), and the overlapping area between the specificity curves of Np ELISA vs. RBD ELISA (Panel B).

commercial test but at a fraction of the cost. Thus, they may be alternative tools for conducting large epidemiological studies in resource limited environments, such as developing countries, when qualify personnel it is available. In this line, the in-house ELISAs require more qualified personnel than the commercial kit, in addition to be more time consuming. In the present study, labor was not considered in the cost comparison. Despite the latter, it is important to note that full-time technicians are commonly present at university laboratories, where a more frequent problem, in developing countries, is the lack of resources to buy reagents rather than labor availability. In this way, the in-house ELISAs is available alternative to conduct COVID-19 surveillance programs based at university/research institutions.

The OD cut-off optimization and performance estimates considered local epidemiological conditions; thus, study results represent the expected performance of the assays in the target population. ROC analysis indicated a non-significant higher sensitivity and statistically significant lower specificity of immunoglobins targeting RBD in comparison to Np. In opposition,

**Table 1. Cross-tabulation of in-house ELISAs (Np & RBD ELISA) results and a commercial lateral flow chromatographic immunoassay (LFCI) result.**

|  |  | LFCI[a] |  |
|---|---|---|---|
|  |  | Positive | Negative |
| **Np ELISA** | **Positive** | 44 | 2 |
|  | **Negative** | 3 | 45 |
|  |  | Positive | Negative |
| **RBD ELISA** | **Positive** | 44 | 7 |
|  | **Negative** | 3 | 40 |

[a]The Diagnostic Kit for IgM / IgG Antibody to Coronavirus (SARS-CoV-2) (Lateral Flow), LIVZON, China

**Table 2. Cross-tabulation of SARS-CoV-2 infection status and a commercial lateral flow chromatographic immunoassay (LFCI) test result.**

|  |  | Disease Status | |
|---|---|---|---|
|  |  | Confirmed | Control |
| LFCI[a] | Positive | 47 | 0 |
|  | Negative | 3 | 44 |

[a]The Diagnostic Kit for IgM / IgG Antibody to Coronavirus (SARS-CoV-2) (Lateral Flow), LIVZON, China

previous studies have shown a higher Np sensitivity than RBD, and higher specificity of RBD than Np [24, 25]. Even though these authors argue that this phenomenon occurs in the early phase of infection, with IgG as the main isotype. In this line, Burbelo et al. [19] showed in a longitudinal study that antibodies against Np emerged before antibodies against the S protein. In the present study, confirmed COVID-19 cases had manifested clinical signs before serum collection (median = 11.5 days, IQR: 9–15 days). Thus, sensitivity and specificity results reflect the performance of Np and RBD antigens at later stages of the infection under local conditions. Moreover, considering that the intended use of the in-house ELISA assays is for surveillance purposes, study outputs could represent the expected performance in the general population, for the detection of past infections of SARS-CoV-2. A recent meta-analysis study [21], assessing the performance of serological tests for the detection of antibodies against SARS-CoV-2, has shown that ELISA tests (detecting IgG or IgM) present lower sensitivity than the assays evaluated in this study, where on average those assays presented a sensitivity of 84.3% (95% CI: 75.6–90.9%). On the other hand, that meta-analysis showed a comparable specificity (range 96.6–99.7%), at least for Np ELISA. In this sense, the combined capture of three immunoglobulins, and particularly the inclusion of IgA, could contribute to explaining the higher sensitivity observed. Similarly, Ma et al. [30] reported that measuring IgA in addition to IgG or IgM would increase the sensitivity of the assay. In this line, previous reports have indicated that SARS-CoV-2 elicits robust humoral immune responses, including the production of IgM, IgG, and IgA immunoglobulin isotypes [34]. Patients have been shown to achieve seroconversion and produce detectable antibodies within 20 days from the symptom onset, though the kinetics of IgM and IgG production was variable [6, 35, 36]. A serological screening that includes the detection of IgA, IgM and IgG could be more consistent as a strategy to prevent the spread of the virus, given the need to maximize the sensitivity [15], when the diagnostic assay is used as a screening test.

Based on the results of the Np and RBD in-house ELISAs, the combined detection of three immunoglobulin isotypes (IgA, IgM and IgG) could be more sensitive than assays that only detect one or two immunoglobulin isotypes, particularly for screening purposes or in studies to evaluate the exposition levels in the general population to SARS-CoV-2 virus, regardless of stage, severity, and symptoms of COVID-19 disease. However, the increase in sensitivity is offset by the reduction in specificity, which was especially evident when the RBD antigen was used. This could be also appreciated when contrasting OD distributions for both antigens (Fig 2), where the comparison between control sera showed that RBD ELISA tend to have higher OD values than Np ELISA in this group, indicating a lower specificity of this antigen.

## Conclusion

Considering the intended purpose of the in-house ELISAs, a relevant element was to obtain accurate sensitivity and specificity estimates, to support further epidemiological studies. In this sense, estimates for Np ELISA presented a variation (in relation to the mean) between 8–16%,

whereas RBD ELISA presented a variation between 14–21%. Thus, relatively precise estimates were obtained. Moreover, at the comparison with the commercial LFCI test, the Np ELISA presented a comparable performance to the LFCI test, while the RBD ELISA presented a significant lower specificity than the LFCI test and the Np ELISA. Therefore, it can be concluded that the Np ELISA is a better assay than RBD ELISA, which presented performance comparable to a commonly used commercial test, but at a quarter of the cost (when labor was not considered), making it a viable alternative for surveillance studies in developing countries under specific circumstances. Particularly, for programs based of university/research institutions, which commonly count with qualified technicians and equipment, thus analysis cost would be primarily spent on reagents.

## Supporting information

**S1 Table. In-house ELISAs reagents and instruments.** List of all reagents and instruments (brand and source) used to run both in-house ELISAs.
(PDF)

**S2 Table. Cost of in-house ELISA reagents.** Bulk cost of all reagents used to run both in-house ELISAs. Figures have been converted from Chilean Pesos (CLP) to US dollar (USD) using exchange rate of July 2020, that correspond to the date when regents were bought.
(PDF)

**S3 Table. In-house ELISAs and rapid test results.** Anonymized individual results for Np ELISA, RBD ELISA, and the lateral flow chromatographic immunoassay (LFCI) test kit (The Diagnostic Kit for IgM / IgG Antibody to Coronavirus (SARS-CoV-2) (Lateral Flow), LIV-ZON, China).
(XLSX)

## Acknowledgments

A special thanks to Dr. Ratto (Animal Science Institute, UACh) for the facilities granted for access and the use of the photo-documentation machine during the COViD-19 pandemic and to the Graduate School of the Faculty of Veterinary Sciences (UACh) for co-financing the acquisition of the plate reader (HR801, Shenzhen Highcreation Technology Co., Ltd). We also thank the SEREMI Coquimbo region, Servicio de Salud de Coquimbo, Hospital Regional de Ovalle, and Municipality of Monte Patria for their logistical support, and Vinka Valencia, Paula Rojas, Cristian Brevis and Paxelia Huertas for their invaluable assistance during field work for the pre-pandemic sampling. Pre-pandemic sampling was supported by EU-LAC Health (EULAC/FONIS T020067), through the PERITAS (Molecular epidemiological studies on pathways of transmission and long-lasting capacity building to prevent cystic echinococcosis infection) project (G.A. EULACH16/T02-0067; https://www.iss.it/en/web/iss-en/who-cc-peritas).

## Author Contributions

**Conceptualization:** Franz Villarroel-Espíndola, Maritza Navarrete, Alberto Fica, Gerardo Acosta-Jamett, Cristóbal C. Verdugo.

**Data curation:** Alfredo Ramírez-Reveco, Gerardo Velásquez, Christopher Aros, Gabriela Navarrete, Natalia Castro, Cristóbal C. Verdugo.

**Formal analysis:** Alfredo Ramírez-Reveco, Gerardo Velásquez, Christopher Aros, Gabriela Navarrete, Anita Plaza, Cristóbal C. Verdugo.

**Funding acquisition:** Cristóbal C. Verdugo.

**Investigation:** Maritza Navarrete.

**Methodology:** Alfredo Ramírez-Reveco, Maritza Navarrete, Alberto Fica, Gerardo Acosta-Jamett, Cristóbal C. Verdugo.

**Project administration:** Natalia Castro, Cristóbal C. Verdugo.

**Supervision:** Maritza Navarrete, Natalia Castro, Cristóbal C. Verdugo.

**Writing – original draft:** Alfredo Ramírez-Reveco, Claudio Verdugo, Gerardo Acosta-Jamett, Cristóbal C. Verdugo.

**Writing – review & editing:** Franz Villarroel-Espíndola, Maritza Navarrete, Cristóbal C. Verdugo.

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
