## [Decision Letter · Decision Letter 0]

8 Aug 2022

PONE-D-22-16590Performance estimation of two in-house ELISA assays for COVID-19 surveillance through the combined detection of anti-SARS-CoV-2 IgA, IgM, and IgG immunoglobulin isotypesPLOS ONE

Dear Dr. Verdugo,

Thank you for submitting your manuscript to PLOS ONE. After careful consideration, we feel that it has merit but does not fully meet PLOS ONE’s publication criteria as it currently stands. Therefore, we invite you to submit a revised version of the manuscript that addresses the points raised during the review process.

The introduction is scanty and does not give adequate background of the prevalence of SARS-COV-2 in Chile.The author must clarify how the participant number was calculated to achieve satisfactory statistical power for appropriate interpretation of results. A sample size of 94 include both cases and controls is too small to use in a disease which is pandemic.Please provide additional details regarding participant consent. Please ensure that you have specified what type you obtained (for instance, written or verbal, and if verbal, how it was documented and witnessed). If your study included minors, state whether you obtained consent from parents or guardians.The rapid test used does not include IgA, and it is not clear how the results for the 2 in-house assay were compared with the rapid while it does not contain similar antibodies.No comparison has been performed with other formal serological assay that are either based on the ELISA or the chemiluminescent detection principle, but just a comparison with a lateral flow assay is reported which is not sufficient.The materials and methods section is poor in experimental details. I suggest, to allow a better to understanding to re-write this section.Principle and protocol of the rapid assay is not reported in the materials and methods section. Please, the authors provide it.Please list the names, sources of all reagents, software used in the methodology.Please list all the instrument used in the methodology eg. What was used to achieve a 37^0^C temperature.Line 145 should read Study design and laboratory analysis not analyses.Line 238: ‘’Figure 3B” explanation in the text does not correspond to Figure 3B/ Figure 2 B is not explained.All the abbreviations must be written in full before abbreviation.The author recommended the use of in house assay over lateral flow assay, however, several study has shown that lateral flow assay are cheap and easy to use especially in developing countries where there is a lack of infrastructure and their maintenance. Where available the infrastructure is available in university settings as mentioned in by the authors but may not be available in and as infrastructure and maintenance, that may be limited in clinics which may be remote from university laboratories.  Include a table with the cost of all reagents used in the in-house to justify the use of in house as an alternative diagnostic tool. Please submit your revised manuscript by Sep 22 2022 11:59PM. If you will need more time than this to complete your revisions, please reply to this message or contact the journal office at plosone@plos.org. Please include the following items when submitting your revised manuscript:A rebuttal letter that responds to each point raised by the academic editor and reviewer(s). You should upload this letter as a separate file labeled 'Response to Reviewers'.A marked-up copy of your manuscript that highlights changes made to the original version. You should upload this as a separate file labeled 'Revised Manuscript with Track Changes'.An unmarked version of your revised paper without tracked changes. You should upload this as a separate file labeled 'Manuscript'.

We look forward to receiving your revised manuscript.

Kind regards,

Maemu Petronella Gededzha, Ph.D

Academic Editor

PLOS ONE

Journal Requirements:

“Pre-pandemic sampling was supported by EU-LAC Health (EULAC/FONIS T020067), through the PERITAS (Molecular epidemiological studies on pathways of transmission and long-lasting capacity building to prevent cystic echinococcosis infection) project (G.A. EULACH16/T02-0067; https://www.iss.it/en/web/iss-en/who-cc-peritas).”

“- CrV

- ANID COVID 0585

- Agencia Nacional de Investigacion y Desarrollo del Ministerio de Ciencias de Chile

- www.anid.cl

5. Please amend the manuscript submission data (via Edit Submission) to include author Gerardo Velázquez, Christopher Aros, Gabriela Navarrete, Franz Villarroel-Espindola, Maritza Navarrete and Natalia Castro

Reviewers' comments:

Reviewer's Responses to Questions

**Comments to the Author**

1. Is the manuscript technically sound, and do the data support the conclusions?

Reviewer #1: Yes

Reviewer #2: Yes

2. Has the statistical analysis been performed appropriately and rigorously? 

Reviewer #1: Yes

Reviewer #2: Yes

3. Have the authors made all data underlying the findings in their manuscript fully available?

Reviewer #1: Yes

Reviewer #2: No

4. Is the manuscript presented in an intelligible fashion and written in standard English?

Reviewer #1: Yes

Reviewer #2: Yes

5. Review Comments to the Author

Reviewer #1: The manuscript was well written and the Authors were able to show that the two in-house ELISA assays were efficient and could be a vital tool in the surviellance studies of COVID-19, However, the study showed that using LFCI assay is still the best assay, although its is a bit expensive for poor resource limited settings, So is the Author saying that its better to use LFCI assay than the other two in-house tested assay they displayed? or what are they actually recommending?

Please correct these minor findings on the manuscript listed below:

Abstract

Line 34, Np and RBD are abbreviated and no written in full, and it’s the first time they are being mentioned, for someone who doesn’t know COVI-19, they might not understand what they stand for.

Introduction

Line76, RT-qPCR mentioned for the first time, write in full then abbreviate.

Materials and Methods,

Line 162, Is it supposed to be 1XPBS, -Tween20 0.05% in brackets or….. ?. the same query applies for Line 168

Line 165 and line 172, was it supposed to be a reference added after as “previously described by” .

Line 173, what does TMB stand for? First time mention and abbreviated.

Line 473, in the text body it was receptor binding domain, now its region binding domine, is there a difference or just some typing error.

Line 476, If its respectively isn’t it supposed to be written starting with the light orange and in red.

Line 475, in the body text, RBD was being described as the receptor binding domain, now on the figure legend its written as the region binding domine, is it still one and the same thing or there is a difference between the two?

Reviewer #2: The manuscript reports on a good advancement especially for low and middle income countries that may not have enough funds to commission population based surveillance due to cost of commercial kits. A few typographical and grammatic errors were noted.

Line 31, Include free 'of' SARS-CoV-2...

Line 49, replace the word 'Than' with 'to'...

Line 145 and 150, replace "analyzes with 'analysis'

Line 151, delete the word 'in'

Line 324, replace word 'than' with "to"

6. PLOS authors have the option to publish the peer review history of their article (what does this mean?). If published, this will include your full peer review and any attached files.

Reviewer #1: No

Reviewer #2: No

---

## [Author Response · Author response to Decision Letter 0]

17 Oct 2022

Response to Editor comments

2. The introduction is scanty and does not give adequate background of the prevalence of SARS-COV-2 in Chile.

Introduction has been edited including a general description of the epidemic development and government response in the country lines 58-63. Despite this, we would like to highlight that a proper prevalence estimate (base on a random selection in the general population) is the outcome of a following paper from the present manuscript that will be comparing official diagnostic rates versus true prevalence estimating (base on the Se & Sp performance of the in-house ELISA test).

3. The author must clarify how the participant number was calculated to achieve satisfactory statistical power for appropriate interpretation of results. A sample size of 94 include both cases and controls is too small to use in a disease which is pandemic.

Respectfully, we do not agree with the opinion of the Reviewer. Sample size calculation (method & parameters) were stated in lines 196-198 of the original submitted manuscript. In particular, we followed a sample size calculation specific for the ROC analysis (Obuchowski, NA. 2005), which is the main analysis of the paper (lines 120-122). Thus, sample size calculation was in line with the study objective, the intended use of the test, and the statistical analysis used in this research. For clarity, sample size calculation information has been moved from the “Statistical analysis” (previous manuscript version) to the “Study population and sample collection” section (lines 119-124). 

4. Please provide additional details regarding participant consent. Please ensure that you have specified what type you obtained (for instance, written or verbal, and if verbal, how it was documented and witnessed). If your study included minors, state whether you obtained consent from parents or guardians.

This section has been edited (lines 144-158).

5. The rapid test used does not include IgA, and it is not clear how the results for the 2 in-house assay were compared with the rapid while it does not contain similar antibodies.

As the Reviewer points out, the commercial rapid test used includes only detections of IgG and IgM and the in-house ELISA evaluated in this study we use the total detection of immunoglobulins (IgA, IgM and IgG). Despite of this, the comparison was carried out in terms of the final assessment that each tool provided. Specifically, in term of their dichotomic outcome (positive/negative). This is a very standard approach in the epidemiology, where the agreement between two tests can be assessed by a 2-by-2 table and computing Coheen’ Kappa based on the final-outcome despite having different chemistry or targets (Methods in Epidemiologic Research, 2021 ISBN-13: 978-0919013735). For clarity, we have edited the statistical section (lines 227-233).

We would light to highlight that we performed a diagnostic Se & Sp estimation rather than an analytical Se & Sp estimation, which allows for the end-result comparison of two different tests, such the one conducted in the present research. Additionally, the potential effects (Se and Sp) of the presence/absence of IgA, between both tests, were discussed in lines 346-360. 

6. No comparison has been performed with other formal serological assay that are either based on the ELISA or the chemiluminescent detection principle, but just a comparison with a lateral flow assay is reported which is not sufficient.

We followed the STARD guidelines, which is an international effort of researchers to standardize test validation reports since 2003 (updated in 2015) (https://www.equator-network.org/reporting-guidelines/stard/). In such internationally recognized guidelines, there is no such thing as a minimum (more than one) of extra tests for comparison purposes. Independently of this, the main objective of this research (lines 110-115) was achieved through the ROC analysis. Specifically, OD cut-off, sensitivity and specificity estimates were derived directly from the ROC curves (lines 118-124 & 215-224).

In this way, the comparison of the in-house ELISA tests with LFA was a complementary analysis, which was carried out with the purpose of having a reference with the most widely used serologic test in the country. Therefore, in the light of the internationally accepted standard guidelines for test validation, in addition to the study objective, we (respectfully) do not agree with the Reviewer comment that we have not performed enough analysis to address the purpose of this research. 

7. The materials and methods section is poor in experimental details. I suggest, to allow a better to understanding to re-write this section

Materials and methods sections has been updated 

8. Principle and protocol of the rapid assay is not reported in the materials and methods section. Please, the authors provide it.

The following text has been included:

Serum samples for rapid test were processed according to the manufacturer's instructions (Zhuhai Livzon Diagnostics Inc.). Briefly: 10 μL of serum sample were added into the sample well of both IgM test and IgG test cassettes, then 2 drops (about 100 μL) of sample diluent were added vertically. Finally, within 1 to 15 minutes after sample addition, the results were interpreted as positive when both the control line and the test line appeared. The results were interpretated as negative when only control line appeared.

9. Please list the names, sources of all reagents, software used in the methodology.

Information requested not explicitly presented in the M&M section has been included as a supplementary material (S1 Table). All software used were already informed in the statistical analysis section (lines 215-216, 227).

10. Please list all the instrument used in the methodology eg. What was used to achieve a 370C temperature.

Included as a supplementary material (S1 Table).

11. Line 145 should read Study design and laboratory analysis not analyses.

Updated as requested

12. Line 238: ‘’Figure 3B” explanation in the text does not correspond to Figure 3B/ Figure 2 B is not explained.

The explanation in the text indicated by the reviewer corresponds to Figure 2B, and not to 3B, this was corrected in the text

13. All the abbreviations must be written in full before abbreviation.

Updated as requested

14. The author recommended the use of in house assay over lateral flow assay, however, several study has shown that lateral flow assay are cheap and easy to use especially in developing countries where there is a lack of infrastructure and their maintenance. Where available the infrastructure is available in university settings as mentioned in by the authors but may not be available in and as infrastructure and maintenance, that may be limited in clinics which may be remote from university laboratories. Include a table with the cost of all reagents used in the in-house to justify the use of in house as an alternative diagnostic tool. 

We have edited the discussion and conclusion of the manuscript in order to provide a clearer message to the readers (lines 49-50, 308, 325, 330-331, and 381-385). In particular, we provide a more balanced conclusion in which scenario the in-house ELISA is more convenient to use. Additionally, we have included as supporting information (S2 Table) the cost of the regents used.

---

## [Editor Report · Decision Letter 1]

7 Dec 2022

Performance estimation of two in-house ELISA assays for COVID-19 surveillance through the combined detection of anti-SARS-CoV-2 IgA, IgM, and IgG immunoglobulin isotypes

PONE-D-22-16590R1

Dear Dr. Verdugo,

We’re pleased to inform you that your manuscript has been judged scientifically suitable for publication and will be formally accepted for publication once it meets all outstanding technical requirements.

Kind regards,

Maemu Petronella Gededzha, Ph.D

Academic Editor

PLOS ONE
---

## [Editor Report · Acceptance letter]

27 Jan 2023

PONE-D-22-16590R1 

Performance estimation of two in-house ELISA assays for COVID-19 surveillance through the combined detection of anti-SARS-CoV-2 IgA, IgM, and IgG immunoglobulin isotypes 

Dear Dr. Verdugo:

I'm pleased to inform you that your manuscript has been deemed suitable for publication in PLOS ONE. Congratulations! Your manuscript is now with our production department. 

Kind regards, 

on behalf of

Dr. Maemu Petronella Gededzha 

Academic Editor

PLOS ONE